# Robust Depth Estimation for Light Field Microscopy

**DOI:** 10.3390/s19030500

**Published:** 2019-01-25

**Authors:** Luca Palmieri, Gabriele Scrofani, Nicolò Incardona, Genaro Saavedra, Manuel Martínez-Corral, Reinhard Koch

**Affiliations:** 1Department of Computer Science, Christian-Albrecht-University, 24118 Kiel, Germany; rk@informatik.uni-kiel.de; 2Department of Optics, University of Valencia, E-46100 Burjassot, Spain; gabriele.scrofani@uv.es (G.S.); nicolo.incardona@uv.es (N.I.); genaro.saavedra@uv.es (G.S.); manuel.martinez@uv.es (M.M.-C.)

**Keywords:** depth estimation, light field, microscope, stereo matching, defocus

## Abstract

Light field technologies have seen a rise in recent years and microscopy is a field where such technology has had a deep impact. The possibility to provide spatial and angular information at the same time and in a single shot brings several advantages and allows for new applications. A common goal in these applications is the calculation of a depth map to reconstruct the three-dimensional geometry of the scene. Many approaches are applicable, but most of them cannot achieve high accuracy because of the nature of such images: biological samples are usually poor in features and do not exhibit sharp colors like natural scene. Due to such conditions, standard approaches result in noisy depth maps. In this work, a robust approach is proposed where accurate depth maps can be produced exploiting the information recorded in the light field, in particular, images produced with Fourier integral Microscope. The proposed approach can be divided into three main parts. Initially, it creates two cost volumes using different focal cues, namely correspondences and defocus. Secondly, it applies filtering methods that exploit multi-scale and super-pixels cost aggregation to reduce noise and enhance the accuracy. Finally, it merges the two cost volumes and extracts a depth map through multi-label optimization.

## 1. Introduction

Light field microscopy was first introduced at Stanford in 2006 [1], and later improved in the same laboratory [2,3,4]. It consists of placing a microlens array (MLA) at the image plane of a conventional microscope, allowing for the capture of light field that records simultaneously both angular and spatial information of microscopic samples.

The main limitation of light field in microscopy is the spatial resolution [5]. To overcome this problem, a change of paradigm was necessary, so that the MLA is set, not at the image plane, but at the Fourier plane [6,7]. This realization of light field concept was named as Fourier integral Microscopy (FiMic). Light field microscopy has been used for several applications, such as brain imaging of neural activities in [8,9,10].

A common goal in microscopy is to estimate the three-dimensional structure of the observed sample. Industrial solutions reach a high accuracy of the reconstruction using different techniques as confocal microscopy [11], interferometry or variational focus [12], scanning electron microscopy (SEM) [13], optical profilometry [14,15], or stereo cameras [16].

However, these methods also present disadvantages in terms of real-time feasibility, sample preparation and costs. Light field concept provides a simpler and inexpensive approach that allows for many applications in real-time. On the other hand, because of the lack of texture and the presence of repetitive patterns that characterize microscopic samples, the task of extracting a depth map from such light fields presents many challenges.

To address these challenges, different methods have been proposed, whose main limitation still is the final resolution: in [17] optical flow and triangular meshes are used, and in [18] a Lytro consumer camera is used to build a light field microscope and a variational multi-scale optical flow algorithm is used to estimate the depth. An interesting approach to estimate depth of thin structure has been proposed in [19], obtaining high quality results at the price of targeting only a small subset of biological images.

Depth estimation from light field images has already been largely studied and different approaches were proposed, using epipolar plane images [20,21], angular or spatial information [22,23], focal stack [24] and correspondences cues combined [25], explicitly modeled occlusion-aware approaches [26], robust pseudo random field [27] and learning based costs [28].

In previous works we extended similar applications to the case of focused plenoptic cameras (plenoptic 2.0) using MLA with a large number of microlenses, where the raw image contains micro-images capturing a portion of the scene. The main targets were the lens selection process for such MLA-based cameras [29] and evaluating different methods for depth estimation [30].

However, FiMic images constitute another special case. Because of MLA position and structure, the light field is sampled differently. It samples the light field as a conventional plenoptic camera (plenoptic 1.0), while at the same time the perspective views are arranged on a hexagonal grid and exhibit large disparities between them, thus an interpolation to transform them into a rectangular grid would produce strong artifacts.

The main contribution of our work is the creation of a framework where different methods are unified and adapted to obtain a more robust and accurate approach. It takes the above-mentioned conditions into account and provides a method for recovering accurate depth information using a sparser light field, i.e., lower number of views with higher disparity shift. The method can be applied not only to FiMic, but also to images acquired with conventional light field cameras, by choosing only a subset of the perspective views, or to sparser light fields.

The structure of the rest of the paper is as follows: in Section 2, the Fourier Integral Microscope will be described; in Section 3, the depth estimation workflow is explained; in Section 4, a Comparative Performance Analysis to prove the quality of the proposed method is performed; then in Section 5, a potential application is shown to enhance the importance of the contributions; and finally in Section 6, a brief summary of the proposed work is given.

## 2. Fourier Integral Microscope

The presented work takes as input light field microscopy images. The light field microscope used here is the FiMic described in [7]. In the FiMic design shown in Figure 1 the MLA is conjugated with the aperture stop (AS) of the microscope objective (MO). In this way, the sensor, that is placed at the focal plane of the MLA, captures the perspective views directly.

Changing the focal length of the two lenses L1 and L2, the designer can change the number of microlenses that fit in the diagonal of the AS. This affects the resolution limit (*r*) and the depth of field (DOF) as follows [7]:(1)$$r=M_{n}\frac{\lambda }{2NA}$$
(2)$$DOF=\frac{5}{4}\frac{\lambda }{{NA}^{2}}M_{n}^{2}.$$

In these equations λ is the wavelength of the incoming light, NA is the numerical aperture of the MO and $M_{n}$ is the number of microlenses that fit in the diameter of the AS. Note that usually the resolution capacity of an optical system is evaluated in terms of $r^{-1}$. It must be underlined that increasing $M_{n}$ has two effects; a decrease of the resolution capacity and an increase of the DOF. An optimal setup for 3D microscopy aims to reach the highest resolution and the largest DOF, but as previously stated, these two factors are inversely affected by $M_{n}$. Therefore, depending on the sample to be reconstructed, $M_{n}$ will be chosen according to the resolution and DOF required.

A typical example of an image acquired with this microscope can be seen in Figure 2, which includes seven elemental images, i.e., perspective views. These images can be seen as a particular case of a multi-view stereo imaging system, where different viewpoints are arranged on an hexagonal grid, having images aligned along at least three epipolar lines. This makes it suitable for correspondence matching.

Moreover, from such views a focal stack of refocused images can be extracted, by overlapping to the central one a shifted version of the elemental images, where the shift has to be oriented towards the center and will be the measure of the depth of the focal plane obtained. The depth of refocusing ($z_{R}$), with respect to the focal plane of the MO (considered as $z_{R}=0\;\mu m$) can be calculated with the following formula [7]:(3)$$z_{R}=s\frac{f_{MO}^{2}}{f_{L}}{\left(\frac{f_{2}}{f_{1}}\right)}^{2}\frac{\delta }{p}.$$

Here $f_{MO}$, $f_{L}$, $f_{2}$ and $f_{1}$ are respectively the focal lengths of the MO, MLA, L2 and L1. Besides, δ is the pixel’s pitch, *p* is the MLA’s pitch, and *s* is the integer number of shifted pixels applied to the focal stack. The use of defocus cue is then directly applicable to the generated focal stack. Due to these basic considerations, our approach aims to create a depth map by combining these two types of vision cues that are suitable for light field images.

## 3. Depth Map Calculation

The presented work builds on several successful ideas proposed for the depth estimation. The core consists in combining different visual cues, namely focus and correspondences, as in [25], to create a more versatile method, but it differs in the depth estimation.

The main contribution of this section is the combination of existing ideas and novel implementations. As it is possible to see in Figure 3, the depth estimation process can be divided into intermediate steps that contributes to the outcome of the algorithm. To achieve high quality results without losing in robustness and flexibility, we designed the estimation process to allow full controls over the parameters and fine-tuning for each single step (see Appendix A).

The pipeline consists in several steps: first, two cost volume cubes are calculated using respectively the elemental images, in Section 3.2, and the focal stack, in Section 3.3. These two cubes are then refined using a multi-scale approach similar to [31] in Section 3.4 and a contribution from superpixels inspired from [32] in Section 3.5. In Appendix A an overview of the parameters is given in Table A1.

Moreover, we introduce additional steps to increase the robustness of the algorithm and to adapt to a vast variety of input images. In Section 3.1 priors are incorporated in the fusion of the data under three different forms. A matting mask is applied to address the dark regions of the scenes, appearing mostly in transparent biological samples, where the empty areas do not capture light. Areas with high and low frequencies are weighted differently on high or low resolution depth estimation, and a failure prediction map is used to weight the contributions of the stereo matching along different epipolar lines.

The next step consists in fusing the two cost volumes and extracting a depth map. This is approached in Section 3.6 as an energy minimization problem, building an energy function that combines both cost volumes and finding the minimum using a multi-label optimization framework.

Finally, we used a post-processing filter to improve the estimation and obtain a smoother depth map. We applied a weighted median filter with integer weights followed by a guided filter, using in both cases the central image as a reference image.

### 3.1. Priors Information

Priors information have been incorporated into the depth estimation pipeline to deal with different kinds of input images, increasing the robustness and the reliability of the proposed approach.

#### 3.1.1. Frequency Mapping

As explored in the literature in [33,34], the best improvements for the multi-scale approach are visible across areas with different frequencies: in high-frequencies areas high-resolution images can obtain the best results, while in low-textured and low-frequency areas a lower resolution leads towards more robust estimation.

Based on this consideration, we built a frequency map image using the difference of gaussians. The algorithm, calculating the difference in the amount of blur between two different blurred versions of the same image, is able to compute a frequency value for each pixel of the reference image.
(4)$$F_{map}=I*\left(g\left(\sigma_{1}\right)-g\left(\sigma_{2}\right)\right)$$
where the reference image is denoted as *I* and $g(\sigma_{1})$, $g(\sigma_{1})$ are gaussian kernels of variance $\sigma_{1}$ and $\sigma_{2}$. To obtain the desired result, we must ensure $\sigma_{1}\neq \sigma 2$. The pixel values will then be use to weight the sum of the depth estimation at different scales, by quantizing the images into $F_{s}$ levels, where each level corresponds to one scaled version.

#### 3.1.2. Failure Prediction Map

In the stereo matching case, the correspondences search is conducted along the epipolar line: this inevitably leads to error in estimation of structures along that line, as pointed out in [35].

We built three different sobel-like kernel filters to detect edges at respectively 0,60 and 120 degrees along the three epipolar lines of the hexagonal grid structures. By convolving the image with such a filter we obtain another weight map that favours the contribution of the estimation coming from the most appropriate direction. In Figure 4 it is possible to see the three kernels and an example of failure maps.

#### 3.1.3. Matting

Some of the assumptions for natural images are not valid when dealing with microscope images. Because of the nature of the object and fluorescence illumination, some scenes exhibit a uniformly dark colored background.

In such dark and uniformly colored areas, almost every possible approach for depth estimation is doomed to fail. Based on these considerations, we adapt a matting approach to mask the areas we do not want to analyze. The proposed method has two steps: the first is creating a so-called trimap, that consists in heavily quantizing the image to three levels representing respectively foreground, unknown areas and background. This can be done by applying a multi-level thresholding on the focused image.

After this trimap is built, the next step consists in determining if the unknown areas belong to the foreground or background. We do this by applying one of the top performer methods in this areas, the three-layer graph approach in [36]. It introduces a new measure for discriminating pixels by using non-local constraint in the form of a three layer graph model to supplement local constraint, consisting of color line model, forming a quadratic programming that can be solved to obtain the alpha matte. The results are shown in Figure 4.

### 3.2. Cost Volume from Correspondences

A stereo matching is performed to compute a three-dimensional cost volume from the elemental images. The cost function used for this scope is
(5)$$C_{cor}\left(p\right)=\alpha_{c}TAD\left(p\right)+\left(1-\alpha_{c}\right)\xi \left(p\right)$$
where we use the notation C(p) to indicate the cost of a pixel *p*. In this formula the combination of the two terms is controlled by the value $\alpha_{c}\in \left[0,1\right]$. The first term TAD is the truncated sum of absolute difference and is calculated between two images:(6)$$TAD\left(x,y\right)=\sum_{r=-hws}^{+hws}\sum_{q=-hws}^{+hws}min(\tau ,|I_{1}\left(x+r,y+q\right)-I_{2}\left(x+d_{x}+r,y+d_{y}+q\right)\left|\right)w_{\left(x,y,r,q\right)}$$

In this notation we substitute the pixel *p* with its *x* and *y* coordinates within the image. The sum is then computed on a window, indices r,q are used to reach the pixel within the window and hws indicates half of the size of the window. To deal with three different epipolar lines, the disparity can be seen as a vector d=(d,θ) where *d* is the disparity value and θ the direction (θ=0,60,120°) and is divided into its two horizontal and vertical components, respectively $d_{x}=d\cos\left(\theta \right)$ and $d_{y}=d\sin\left(\theta \right)$. τ is the threshold for truncating the difference.

The second term ξ(p) indicates the census transform and is calculated as:(7)$$\xi \left(x,y\right)=\sum_{r=-hws}^{+hws}\sum_{q=-hws}^{+hws}HD(\xi (I_{1}\left(x+r,y+q\right),\xi \left(I_{2}\left(x+d_{x}+r,y+d_{y}+q\right)\right)w_{\left(x,y,r,q\right)}$$
where HD(·) indicates the Hamming difference between the census transform of the window around the pixel and the weights are calculated as described in [37]:(8)$$w_{\left(x,y,r,q\right)}=e^{-\left(\frac{\left|I(x,y)-I(x+r,y+q)\right|}{\sigma_{c}}+\frac{dist(x,y,r,q)}{\sigma_{d}}\right)}$$
where $dist\left(x,y,r,q\right)=\sqrt{r^{2}+q^{2}}$ is the euclidean distance between the central pixel and the considered one and $\sigma_{c}$, $\sigma_{d}$ are the corresponding parameters regulating the contribution of color and distance.

The cost volume is generated as follows: the correspondences search is completed along the three epipolar lines obtaining $\left(N_{EI}-1\right)$ different cost volumes, where $N_{EI}$ is the number of elemental images ($N_{EI}=7$ in our case). To merge the cost volumes, the same slice corresponding to a possible disparity value is taken from each cost volume and combined using a weighted average. The failure map calculated in Section 3.1 contains the weights used.

As described in [38], the cost volume is filtered using a guided image filter that takes the colored image as a reference.

### 3.3. Cost Volume from Defocus

To calculate an accurate defocus map the central image is used as a reference and a difference image for each focal plane is calculated. This allows for a more precise calculation with respect to the defocus response case as defined in [25]: a measure of how much a pixel is in focus at a certain distance.

The difference images are calculated with respect to each focal plane in a similar manner as in the correspondence case:(9)$$C_{def}\left(p\right)=\alpha_{d}TAD\left(p\right)+\left(1-\alpha_{d}\right)NCC\left(p\right)$$
where TAD is defined in Equation (Equation 6) and the linear combination of the two terms is controlled by $\alpha_{d}\in \left[0,1\right]$. The term NCC indicates the normalized cross correlation, calculated as:(10)$$NCC\left(x,y\right)=\sum_{r=-hws}^{+hws}\sum_{q=-hws}^{+hws}\left(\frac{\sigma_{I_{1}I_{2}}^{2}\left(x+r,y+q\right)}{\sigma_{I_{1}}\left(x+r,y+q\right)\sigma_{I_{2}}\left(x+r,y+q\right)}\right)w_{\left(x,y,r,q\right)}$$
where the weights $w_{\left(x,y,r,q\right)}$ are defined in Equation (Equation 8) and $\sigma_{I1I2}$, $\sigma_{I1}$, $\sigma_{I2}$ indicates respectively the joint variance, the variance of the first and of the second image. The cost volume is built by stacking the difference images calculated for each focal plane. The cost volume is also filtered using a guided filter as in [38].

### 3.4. Multi-Scale Approach

Cost volume filtering using a multi-scale approach has shown promising results in refining the cost volume for a higher accuracy of the final depth map [31]. To add consistency and robustness to the proposed cost function, we then adopt a multi-scale approach. Taking inspiration from [31], we build three different layers with a scaling factor s=2, to ensure coherence among images. It has been verified in [31] that building more levels does not significantly improve the final estimation. The cost volume calculated on the smaller scale is then upscaled back and propagated to the initial cost volume.

Differently from [31], however, we do not propagate only the best results of the down-scaled estimation. This idea ensures faster computations, but can lead to larger errors. Instead, we sum the whole cost volume using a weighted average based on the priors information, in this case the frequency map.
(11)$$C_{msc}\left(p\right)=w_{s0}\left(p\right)C\left(p\right)+w_{s1}\left(p\right)C_{s1}\left(p\right)+w_{s2}\left(p\right)C_{s2}\left(p\right)$$

As shown in Equation (Equation 11), the multi-scale cost ($C_{msc}$) is a weighted contribution of the cost computed at different scales, being $C_{s1}$ and $C_{s2}$ respectively the costs computed at scale s1=2 and s2=4. In Equation (Equation 11) C(p) indicates a general cost volume: in our case this is applied to both cost volumes, calculated in Equations (Equation 5) and (Equation 9). From now on they will be denoted as $C_{def,msc}$ and $C_{cor,msc}$.

The weights $w_{si},i=0,1,2$ come from the frequency map, that is quantized into a number of levels matching with the number of scales used, and calculated as:(12)$$w_{si}\left(p\right)=\left\{\begin{array}{cc} \gamma_{1} & \mathrm{if}\;p\in f_{i}\\ \gamma_{2} & \mathrm{if}\;p\in f_{i\pm 1}\\ \gamma_{3} & \mathrm{otherwise} \end{array}\right.$$
where we ensure $1\geq \gamma_{1}\geq \gamma_{2}\geq \gamma_{3}\geq 0$, and typical values are $\gamma_{1}=0.6$, $\gamma_{2}=0.3$ and $\gamma_{3}=0.1$. Here we denote with $w_{si}$ the weight relative to the *i*-th scale, with $\gamma_{si}$ the parameter controlling the weight of *i*-th scale contribution and with $f_{i}$ and $f_{i\pm 1}$ the *i*-th and (*i* ± 1)-th level of the frequency map.

This allows to shape the cost volume based on the characteristics of the pixels, e.g., pixels that belong to high frequency areas will have a cost based on higher resolution and viceversa, pixel from texture-less regions will have a cost built using the lower resolution.

Tuning the parameters allows us to control the impact of the down-scaled version, and by changing them we can obtain more detail-preserving or smoother depth maps. Note that by setting both $\gamma_{1}=\gamma_{2}=\gamma_{3}=0.33$ we obtain a standard multi-scale approach that does not make use of the priors information.

### 3.5. Superpixels

Another technique that reported significant improvement in the cost volume filtering is using superpixels. Superpixels were introduced in [39] and exploited in the depth estimation task [19,32]. The idea behind this consists in grouping pixels with similar characteristics to obtain a more consistent depth estimation. The superpixels are built using two parameters, controlling respectively the approximate size and the similarity between the pixels.

For the depth estimation task there are two main ways of using them: by choosing a small size it can be assumed that the portion of the image corresponding to this superpixel belong to a plane. This allows to compute a single depth value for each superpixel, as done in [19]. This works particularly well for structures that do not exhibit abrupt changes in depth. A different way is shown in [32], where larger size is chosen to allow different depths within a single superpixel. A histogram is built and the best depth estimations are selected and used to filter the cost volume.

Based on these observations we build a flexible approach that takes inspiration from the latter one, but can be restricted to the first. We take the cost volume of the pixels within the superpixels and extract a tentative depth map by extracting the minimum values. From this, a histogram is built, where the highest $N_{p}$ peaks are selected. Then a penalizing function for the labels not being a peak is built:(13)$$z\left(t\right)=\max\left(0,1-\sum_{i=1}^{N_{p}}\frac{t-ind(N_{i})}{\sigma }\right)$$
where *t* loops through the labels (different focal planes or disparity values), the maximum function is used to avoid negative functions, $ind(N_{i})$ indicates the index of the *i*-th peak and σ controls the strictness of the index, i.e., how much a peak is widened to its neighbours. This way of building z(t) ensures z(t)∈[0,1], that provides an easy handling of the superpixels contribution. In fact, we can sum this back to our cost volume using a penalizing factor. We can then write:(14)$$C_{sp}\left(p\right)=C\left(p\right)+\rho c_{sp}\left(p\right)$$
where we call $C_{sp}$ the cost volume updated with the superpixel contribution shown as $c_{sp}\left(p\right)$, while ρ is the factor that controls the impact of the contribution. In Equation (Equation 14) C(p) indicates a general cost volume: in our case this is applied to both $C_{def,msc}$ and $C_{cor,msc}$ to obtain respectively $C_{def,msc,sp}$ and $C_{cor,msc,sp}$. Typical values are ρ=0.2, $N_{p}=3$ and σ=3.

Note that by changing the parameters and ensuring a smaller size of the superpixels, a maximum number of peak $N_{p}=1$ and a small σ, we could obtain a single depth for superpixels, as in [19]. The ρ parameter controls the contribution to the cost volume, with a large value of ρ leading to strongly shape the cost volume for pixels belonging to the same superpixel, and a small value of ρ reducing the cost for having different values within the same superpixel.

### 3.6. Depth Map Extraction

Many approaches are applicable to extract the depth map from the cost volume. Local and semi-global approaches as winner-takes-all (WTA), semi-global matching (SGM) and more global matching (MGM) [40] are shown to be outperformed from global approaches, where the depth map refinement is posed as an energy minimization problem. The general solution is obtained through minimization of an energy function that depends on two terms, one data term accounting for the cost volume and a smoothness to ensure consistency between neighbouring pixels.

The energy minimization problem can be addressed using several approaches: the most used approaches consist in using Markov random field, as in [25] where the two cost volumes are refined based on their confidence, or in [41] where different energy functions are analyzed, using graph cuts as in [23] or belief propagation as in [42], or a combination of the above methods [43].

We have chosen to use the graph cuts method to minimize an energy function defined as:(15)$$E\left(p\right)=E_{data}\left(p\right)+E_{smooth}\left(p\right)$$

Intuitively, the data term should include both cost volumes. We used a strategy that allows to cleverly merge the two cubes. By extracting two tentative depth maps using a winner-takes-all approach, some initial considerations can be made. We mainly encounter two cases: one where both guesses agree on a depth value and one where the cost curves have different shapes. The idea here is to apply different weights in these two situations: in the case where there is a large difference, we assume the pixel is unreliable, being either in a texture-less area or part of a repetitive pattern, thus we choose to stick with the defocus estimation, that is most likely to have a guess similar to the real one. In the second case, we choose to give more weight to the correspondence matching, because it is most likely to be more accurate.

We model this by creating an absolute difference map $M_{ad}=\frac{1}{K}\left|d_{def,wta}-d_{cor,wta}\right|$, where $d_{def,wta}$ and $d_{cor,wta}$ are respectively the tentative depth map from the defocus and correspondence cost volumes and *K* is just a normalization factor, that will be used to weight the two contributions of each slice of the cost volume.

Moreover, we want to exploit the points where the two estimations agree. We thus select the reliable pixel where the difference map has its minimum ($M_{ad}=0$) and for these pixels we penalize the curve with proportion to the distance from the minimum, creating ground control points that are less likely to change their value during the optimization and will serve to distribute the correct value to unreliable neighbours.

The final sum of these contributions can be expressed as
(16)$$E_{data}\left(p\right)=\left(1-M_{ad}\left(p\right)\right)C_{def,msc,sp}\left(p\right)+M_{ad}\left(p\right)C_{cor,msc,sp}\left(p\right)+\beta P_{gcp}\left(p\right)$$
where $C_{def}$ and $C_{cor}$ are respectively the cost volume for defocus and correspondences and $P_{gcp}$ is the penalty function used on the ground control points cost curve to enhance the minimum and β is just a scaling factor. The smoothness term ensures consistency across the four adjacent neighbours and can be expressed as $E_{smooth}\left(p\right)=\sum_{q\in N}\left|l\left(p\right)-l\left(q\right)\right|$ where *N* contains the four neighbours of *p* and l(p) indicates the pixel’s label, i.e., its depth value.

The energy is minimized using a multi-linear optimization as explained in [44,45,46]. The final depth map is filtered using a weighted median filter with integer weights and a guided filter to remove last outliers while preserving structures.

## 4. Comparative Performance Analysis

The algorithm has been planned for images acquired using the FiMic light field microscope [7], where only visual comparison is possible. Therefore, to prove the worth of the proposed work, additional comparisons were made. We created a set of synthetically generated images with its corresponding ground truth, where numerical analysis can be conducted.

We compared state-of-the-art work on light field microscopy on their available images and state-of-the art in depth estimation for both real and synthetic images. Methods for light fields are not directly applicable to our images, therefore a fair comparison could not be made.

### 4.1. State-of-the-Art in Light Field Microscopy

Due to its recent introduction in the microscopy field, there are not many approaches available.

To the best of our knowledge, the best approach in literature belongs to [18], that built a light field microscope using a Lytro first generation camera. The light field in this case consists in 9×9 views of 379×380 pixels, with lateral views exhibiting strong presence of noise. Moreover, the scale of such images, in the order of millimeters, are quite different from the images acquired with FiMic, that can discriminate up to some micrometers.

Despite the image conditions, the extracted depth maps achieve a high accuracy in the reconstruction of the captured objects and maintains consistency in the structure, as shown in Figure 5.

The first image is the head of a daisy, where both approaches reach a satisfying solution, but ours shows a higher level of detail and robustness, visible in the three-dimensional representation.

The second object is an interesting case, where noise, defocus and very low-light condition increases the challenges. Nevertheless, our algorithm is able to reconstruct the structure of the tip of a pencil, that is symmetrical on the y-axis, where the previous approach could not find a solution. This is particularly visible on the lower part of the image.

### 4.2. State-of-the-Art in Depth Map Estimation

To ensure that the proposed approach achieves high quality and accurate results, we also compare against the top performing method for stereo and depth from defocus.

We have chosen to compare with the shape from the focus method, originally published in [47] by taking the best focus measure as analyzed in [48] and a stereo method using neural network to estimate disparity through patches of an image, described in [49]. Unfortunately, we cannot train the network because of lack of dataset, so the pre-trained model is used.

The first set of images, shown in Figure 6, consists of biological samples: the first two images show cotton fibers stained with a fluorescent dye. The hird one shows the head of a zebrafish. The target is excited with a single wavelength laser and is emitting light with longer waveleght (i.e., lower energy). With a bandstop filter the laser’s light is filtered out and the light emitted from the sample is captured by the sensor. The wavelength and the color depends on the fluorescent die of the sample.

Therefore, the image exhibits a dark background, where the matting technique described in Section 3.1.3 can be applied. In this case the FiMic was set in order to have: *r* = 2.2 μm, DOF = 46.4 μm and $Z_{R}$ = 17.5 μm.

As expected, approaches from depth map estimation in natural images fail to follow the thin structure because of the lack of texture in the dark region and the presence of repetitive patterns in the fibers.

To ensure a fair comparison, we also applied our matting results to the estimated depth map, obtaining more consistent results. This confirms the high quality of our approach and the robustness of our algorithm that can tackle different kinds of input images.

This being a special case, we also evaluate the results on a different dataset. The second set of images, shown in Figure 7, consists of images of small opaque electrical components. By using a luminance ring it is possible to illuminate the object avoiding most of the shades. For this experiment the FiMic was built differently so it led to: *r* = 14.5 μm, DOF=1812 μm and $Z_{R}$ = 17.5 μm. Such targets do not need a matting pre-processing step and exhibit structure and texture, being therefore more suitable for standard approaches.

Results on these images lead to some considerations: because of the size of the image, the luminance condition and the reflectance of the metal, the depth map calculations are highly challenging, as proven from results of the shape from focus method [48], that recovers only a very noisy reconstruction. Nevertheless, more sophisticated approaches that incorporate filtering steps, as [49], are able to reach satisfying results, obtaining comparable outcomes.

We acknowledge the lack of ground truth images in this field, therefore we propose a numerical evaluation via synthetic images. In recent works Blender has shown to be suitable for emulating light field behaviour and complex scenes [30,50]. We have chosen to use the Blender engine to simulate realistic images of the fibers as if they were captured using the FiMic, thus obtaining the respective ground truth data. To recreate realistic conditions, a special material was generated to emulate the behaviour of semi-transparent fibers and laser illumination. Such images are shown in Figure 8. By doing so, we also obtain the ground truth that allows for a more detailed analysis. Here we show an example of a synthetically generated image and we recap our results in Table 1.

The algorithm was evaluated computing the error as the absolute value of the difference between the estimated disparity and the real one, using the matting mask computed in our approach to reduce the evaluation to the interesting pixels. Results were averaged over a set of five images with different difficulties and shapes to ensure a fair and consistent comparison. The range of the disparities varies among different images, but is consistent within different approaches.

The error is lower in the proposed approach, as well as the standard deviation, confirming the accuracy and the robustness of the method. The algorithm based on neural network [49] obtains similar performances but still exhibits larger errors, as suggested from the higher variance value, while the shape from focus approach [48] fails in achieving high accuracy as expected, being the most naive approach without post-processing. Nevertheless, it shows that the generated images constitute a challenge for off-the-shelf methods for the disparity estimation and therefore enhance the importance of tackling such challenges.

## 5. Applications

Depth information can also be used for displaying the 3D information of the microscopic objects. With the technique of [51], one of the views and its corresponding depth map can be used to generate an integral image. This image, projected in an integral imaging (InI) monitor, provides a 3D display of the sample.

To generate the integral image, the view and its corresponding depth map are merged into a 3D point cloud. From this point cloud, a set of synthetic views are computationally generated, and finally processed to obtain the final integral image, which will be projected in the InI monitor. An InI monitor is implemented placing a microlens array in front of a pixelated screen: the lenslets of the MLA integrate the rays proceeding from the pixels to reconstruct the 3D scene. This kind of 3D display is autostereoscopic (glasses-free), it allows multiple observers to experience full parallax and overcomes the accommodation-convergence conflict [52].

We implemented the InI monitor through a Samsung SM-T700 and a MLA composed of lenlets having pitch p=1.0 mm and focal length f=3.3 mm (MLA from Fresneltech, model 630). We used as input for the technique of [51] the RGB and depth images of Figure 7. The integral image obtained and its projection in the 3D InI monitor are shown in Figure 9.

In this example the image reference plane is set at the background, so the MLA reconstructs the 3D object as a real image, floating in front of the InI monitor. The observer, through his binocular vision system, perceives the depth of the reconstructed objects. A video recording the InI display is visible at the address https://youtu.be/0fPQtckzc_8, in which the parallax and the depth sensation of the reconstructed 3D object is apparent.

In this video, the object shows increasing parallax as we move from far to closer objects. So the left part of the chip and the solder edges move their position with respect to the point of view, while the right part of the chip is almost fixed. This is because during the capture, the chip was tilted with respect to the microscope, as reflected in the depth map of Figure 7.

This technique is very effective for the visualization of the 3D structure of the microscopic samples and proves the usefulness of our work on depth estimation.

## 6. Summary

The work presented in this paper addresses an important challenge, namely estimating the depth, i.e., the three-dimensional structure of microscopic images. Because of the nature of these images, usually different from natural images, standard approaches may fail. Moreover, due to recent development in the light field technology, light field microscope became a valid alternative for its several applications.

We then use the light field images captured from the recently introduced FiMic [7] that exhibit higher resolution and we showed that the proposed algorithm is capable of accurate reconstruction of challenging scenes, even where previous approaches were failing. These improvements can be helpful for several applications, as presented in the last chapter for the case of lenticular stereoscopic displays, therefore constituting an important contribution for the community.

## Figures and Tables

**Figure 1 sensors-19-00500-f001:**
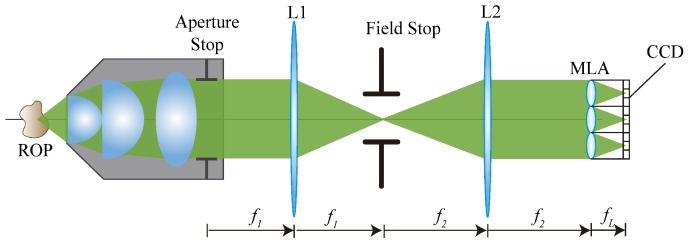
This is the schematic of the FiMic design. From left to right it is possible to distinguish the object, the microscope objective, two lenses (L1 and L2), one field stop, the MLA and the CCD sensor.

**Figure 2 sensors-19-00500-f002:**
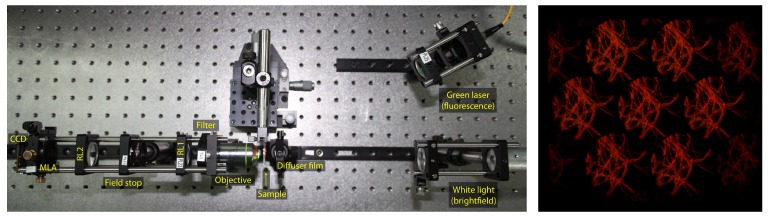
The setup used for the acquisition with the fluorescence laser used to illuminate the samples, and a sample output image acquired with such setup, from where the seven elemental images are visible.

**Figure 3 sensors-19-00500-f003:**
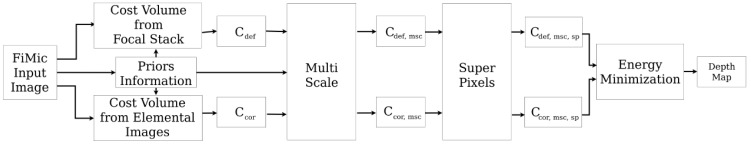
Pipeline of the depth estimation process. The name of cost volume (e.g., $C_{def}$, $C_{cor}$) are consistent with the ones used in the paper.

**Figure 4 sensors-19-00500-f004:**
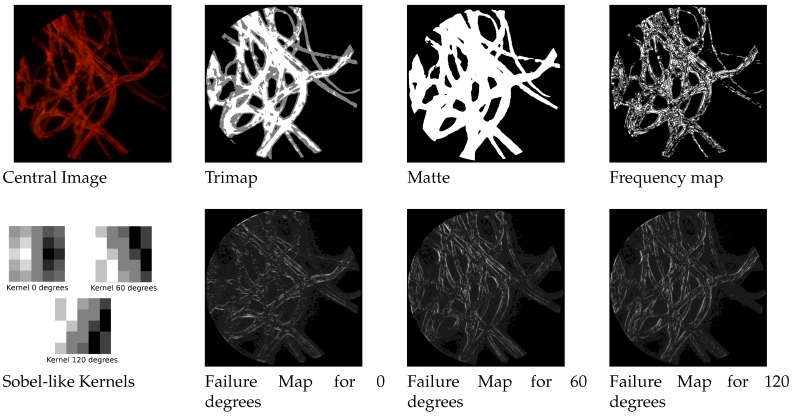
Priors under different form. For coherence, all maps are represented with brighter areas describing higher values. For the failure maps, higher values indicate higher likelihood to fail. In the Sobel-like kernels, bright pixels indicate positive values and dark pixels negative values.

**Figure 5 sensors-19-00500-f005:**
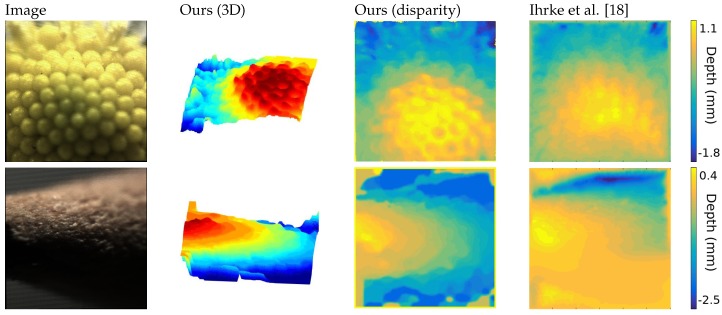
Comparison of depth images from the INRIA dataset. The color coded 3D representation shows the structure of the objects, and the disparity is scaled to try to match the colormap used in [18].

**Figure 6 sensors-19-00500-f006:**
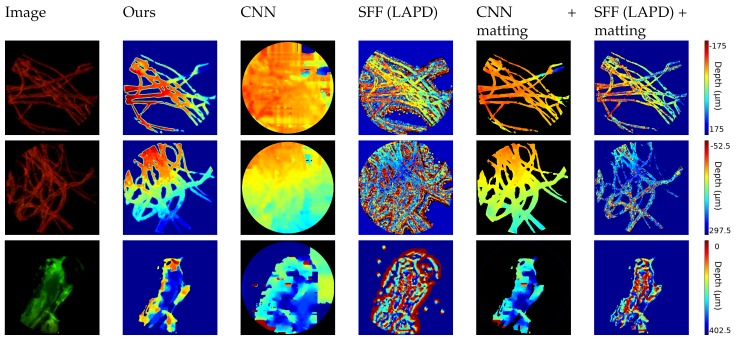
Comparison with Neural Networks (CNN) [49] and Shape from Focus (SFF) [48] on dataset of biological samples: first two rows consist of cotton fibers, last row is the head of a zebrafish.

**Figure 7 sensors-19-00500-f007:**
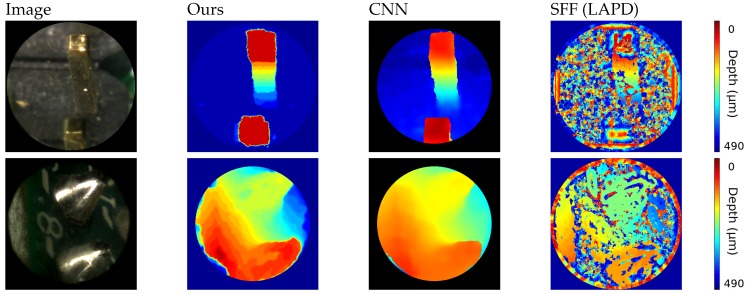
Comparison with CNN [49] and SFF [48] on dataset of opaque electrical components.

**Figure 8 sensors-19-00500-f008:**

Synthetic images generated with Blender. They simulate the behaviour of the FiMic, shown in Figure 2.

**Figure 9 sensors-19-00500-f009:**
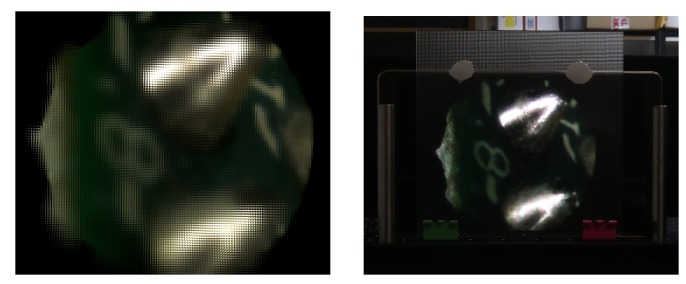
Integral image produced with the method described in [51] and its projection in the 3D monitor.

**Table 1 sensors-19-00500-t001:** Table with results from synthetic images.

Approach	Error	Standard Deviation
Ours	2.32555	1.8154478
CNN [49]	2.4275436	2.4392762
SFF [48]	9.379839	4.1694072

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
