# Peer review of "Robust Depth Estimation for Light Field Microscopy"

_sensors, 2019, doi:10.3390/s19030500_

Round 1

Reviewer 1 Report

The paper describes an approach for estimation of depth in images captured by a Fourier Integral Microscope (FiMic). Since the images of biological samples and other microscopic objects are different that other natural images, the paper describes a depth estimation approach that combines correspondence and depth from defocus techniques along with various contextual priors to arrive at a depth map. The motivation behind this work is clear. While the approach has been designed to particularly handle imaging from the Fourier Integral Microscopy (a recent variation of Light-field Microscopy), it is interesting because it brings together many different pieces from the state of the art to put together a depth estimation approach.

My concerns / questions / comments:

1. Given this is relatively new area, there are not many ways to validate the approach. While authors acknowledge as such in the paper, I have some concerns regarding how the evaluation of the method was done. First, when comparing against Lytro camera, a favorable comparison with [18] is shown, but I wonder how the approach would compare with [19]. Lytro camera seems to be also used in [19], where examples of depth estimation over the order of micrometers are shown. Even if the method in [19] is meant for thin structures, it could be used for comparison in the cotton fibers example shown in Figure 5.

Second issue is related to quantitative results of depth estimation compared to ground truth.  The main issue with simulating the examples using Blender is that it is not clear how the images were generated. More importantly, it is not clear how (and why) these images should be considered representative of the images generated with FiMic. I’m wondering if there could be other ways to generate ground truth data. For example, instead of simulation, why not generate real ground truth by moving a planar object at different depths or using object with known depth?

Third, colorbars from the results are missing, which in the context of this paper are really important.

2. The overall description of the approach is somewhat confusing to me and it could be partly related to sloppy mathematical definitions of some terms. It is claimed that cost volume filtering with guided filter allows achieving higher accuracy (lines 149-153), but there are no details about how this is done or any analysis of its impact on the results. Moreover, many details regarding how the priors are computed and applied within the algorithm are missing (for example, DoG). Is multi-scale approach applied only to the cost volume from correspondence or for both correspondence and defocus? I ask because C(p) term (for correspondence) appears in equation 10, but I don't see the term for defocus (lower case Greek letter delta). In fact, I don't see the term for cost volume defocus appearing anywhere other than in equation 8, although I think it is being used in subsequent steps. In the energy minimization section, we encounter different terms for cost volume for correspondence and defocus in equation 15. Are these same or different that those previously defined? What am I missing here? I suggest a figure that would provide an overview of the approach in terms of intermediate results / block diagram or some combination of these ideas to clarify the approach.

Following up on the same issue of clarity, the first few lines in section 3.3 are not clear to me. What is "defocus response case" as mentioned here? Also, for defocus could you please clarify whether surrounding images from different focal plane are used for computing the difference compared to the center image?

Also, could you please clarify how disparity components are handled for different epipolar lines?

3. There are a large number of parameters defined in the approach but the values for them (or most of them) are not shown or discussed. It would be interesting to know what parameters were chosen, why these values were chosen, and how particular values affect the results. Importantly, different parameters might be important in different applications and this discussion would help understand how the proposed approach can be applied in future cases.

4. Instead of the approach described in the paper for Matting, what would happen if just a simple threshold is used? Visually, it looks like a threshold might work just as good or almost as good. So the point of the question is, how to evaluate the significance of this step compared to plain thresholding?

Minor comments:

In equation 1 and 2 the symbols for numerical aperture and number of microlenses are very confusing (N and NA). Could you please differentiate better between those two? 

Some pages have lines without line numbers at the bottom.

Figure 4 is not referenced in the text.

Author Response

Thank you very much!

Reviewer 2 Report

The paper presents useful methods for using microscope with microlens array. Source codes and data used in the research are openly available. It makes it possible to further develop technologies described in the paper. My concerns about the paper are the following:

1. The abstract describes mainly the topic of light field technologies. However, the abstract should contain more information on the authors' contribution to this topic. It is described only in the last sentence of the abstract starting with "In this work...". I think this should be improved.

2. Authors should clarify what is their original contribution. In the beginning of Sect. 3 there is a sentence: "The presented work builds on several successful ideas proposed for the depth estimation". Did authors contribute to this field of research by selecting these methods and did they tested other methods?

3. Figure 2 shows 7 images. However, the paper is written as if only two of them were considered like in a stereo camera. It is not clear how all these images were processed if they were processed. Moreover, authors compare results with CNN (Zbontar and LeCun [47]) which was designed for stereo cameras.

4. There are no references to equations presented in 3.2 and 3.3. These equations should be also corrected. For example, in Eq. (5) there is q in the range between -hws and +hws. The equation indicates that q is a scalar, not a vector.  In this equation there is also I1(x+q,y+q). This means that q changes simultaneously both for x and y. Such calculations process points located in a straight inclined line instead of processing all points included in a rectangular aggregating window located around x,y.

5. Authors do not refer to their previous papers concerning the same subject (e.g. "The Plenoptic 2.0 Toolbox: Benchmarking of Depth Estimation Methods for MLA-Based Focused Plenoptic Cameras" and "Optimizing the Lens Selection Process for Multi-Focus Plenoptic Cameras and Numerical Evaluation".) The contribution of this paper in comparison to previous papers should be clear.

6. Author Contributions: "The sentence Authorship must be limited to ... " should be deleted.

Author Response

Thank you very much!

Round 2

Reviewer 1 Report

Thank you for your work on addressing initial comments and questions.

I just have a few minor follow-up questions:

1.    Could you please briefly incorporate in the manuscript the explanation for defocus response mentioned in the response to Point 2c (“a measure of how much a certain pixel is in focus at a certain distance”).

2.    The question in Point 2d was actually related to disparity components dx and dy shown in equations 6 and 7. I’m sorry for not making it clear before. It is mentioned in lines 162-163 that dx and dy are horizontal and vertical components of disparity. Given that there are three epipolar lines, could you please further clarify what horizontal and vertical components of disparity refer to?

Author Response

Thank you for your work on addressing initial comments and questions.

Reviewer 2 Report

I think the paper can be accepted in the present form.

Author Response

Thanks.
